# Childhood Malnutrition and Association of Lean Mass with Metabolome and Hormone Profile in Later Life

**DOI:** 10.3390/nu12113593

**Published:** 2020-11-23

**Authors:** Gerard Bryan Gonzales, Natasha Lelijveld, Celine Bourdon, Emmanuel Chimwezi, Moffat J. Nyirenda, Jonathan C. Wells, Marko Kerac, Robert H. J. Bandsma

**Affiliations:** 1Nutrition, Metabolism and Genomics Group, Division of Human Nutrition and Health, Wageningen University, 6708 WE Wageningen, The Netherlands; 2Laboratory of Gastroenterology, Department of Internal Medicine and Paediatrics, Faculty of Medicine and Health Sciences, Ghent University, 9000 Ghent, Belgium; 3Emergency Nutrition Network, Oxford OX5 2DN, UK; natasha.lelijveld.11@alumni.ucl.ac.uk; 4Translational Medicine Program, Hospital for Sick Children, Toronto, ON M5G 1X8, Canada; celine.bourdon@sickkids.ca (C.B.); robert.bandsma@sickkids.ca (R.H.J.B.); 5The Childhood Acute Illness & Nutrition Network, Nairobi 43640-00100, Kenya; 6Department of Nutritional Sciences, Faculty of Medicine, University of Toronto, Toronto, ON M5S, Canada; 7Malawi-Liverpool Wellcome Trust Clinical Research Programme, University of Malawi College of Medicine, Blantyre, Malawi; alinafekc@gmail.com; 8MRC/UVRI Uganda Research Unit, Entebbe, Uganda; moffat.nyirenda@mrcuganda.org; 9Childhood Nutrition Research Centre, Population, Policy and Practice Research and Teaching Department, UCL Great Ormond Street Institute of Child Health, London WC1N 1EH, UK; jonathan.wells@ucl.ac.uk; 10Department of Population Health, London School of Hygiene & Tropical Medicine, London WC1E 7HT, UK; Marko.Kerac@lshtm.ac.uk

**Keywords:** severe malnutrition, non-communicable disease, lean mass, body composition, DOHAD, hormone, bio-impedance

## Abstract

This study aimed to determine the associations of targeted metabolomics and hormone profiles data with lean mass index (LMI), which were estimated using bioelectrical impedance, in survivors of child severe malnutrition (SM) (*n* = 69) and controls (*n* = 77) in Malawi 7 years after being treated. Linear associations between individual metabolite or hormone and LMI were determined, including their interaction with nutrition status 7 years prior. Path analysis was performed to determine structural associations. Lastly, predictive models for LMI were developed using the metabolome and hormone profile by elastic net regularized regression (EN). Metabolites including several lipids, amino acids, and hormones were individually associated (*p* < 0.05 after false discovery rate correction) with LMI. However, plasma FGF21 (Control: β = −0.02, *p* = 0.59; Case: β = −0.14, *p* < 0.001) and tryptophan (Control: β = 0.15, *p* = 0.26; Case: β = 0.70, *p* < 0.001) were associated with LMI among cases but not among controls (both interaction *p*-values < 0.01). Moreover, path analysis revealed that tryptophan mediates the association between child SM and LMI. EN revealed that most predictors of LMI differed between groups, further indicating altered metabolic mechanisms driving lean mass accretion among SM survivors later in life.

## 1. Introduction

Severe malnutrition (SM) in children remains a global health burden that requires urgent attention [1]. High mortality among children in the short term is well documented [2,3]. However, studies on the long-term effects of SM remain limited. While mortality among children with SM has been declining due to improved community and hospital-based management guidelines, concerns are growing about longer-term health outcomes of SM survivors [2]. We have previously shown that survivors of SM have deficient growth, including reduced lean mass compared to children with no history of SM [4]. Recent reports suggest that early postnatal growth failure and SM are associated with a higher risk of developing non-communicable diseases (NCD) during adulthood [5,6,7]. This increased risk is compounded by changing global dietary habits, which are characterized by higher caloric consumption, leading to a double burden of malnutrition [8]. The mechanisms driving this risk are important, but knowledge gaps still remain.

Body composition has a strong influence on NCD risk and can be explained by the capacity-load model [9]. In this conceptual framework, NCD risk is elevated when the metabolic load exceeds metabolic capacity [9]. Lean mass is a good marker for metabolic capacity and is inversely associated with NCD risk [10]. Lower lean mass is also associated with reduced insulin sensitivity [11], increasing the risk for type 2 diabetes, especially in obese individuals [12]. Furthermore, having lower lean mass elevates the risk of other complications, such as adverse cardiovascular events among people with type 2 diabetes [13] and coronary heart disease [14,15]. Mortality among patients with cardiovascular diseases is lowest among those with high lean mass and low fat mass [16,17]. Therefore, strategies to improve lean mass may reduce the risk of NCD development and NCD complications in later life.

Identifying the mechanisms linking childhood malnutrition to long-term NCD development could allow us to identify opportunities for specific interventions to reduce such risk. Therefore, this study aimed to explore factors associated with lean mass deficit in SM survivors by assessing its association with the metabolome and hormonal profile among Malawian children.

## 2. Materials and Methods

### 2.1. Study Design and Setting

We performed a secondary analysis of data obtained from our previous work in Malawi, the ChroSAM cohort [4] (Chronic Disease Outcomes following Severe Malnutrition), where plasma samples were obtained 7 years after being treated for severe malnutrition and previously analyzed for metabolome and hormonal profiling [18]. Plasma samples were also collected and analyzed from controls, consisting of sibling and community controls with no history of severe malnutrition. The original cohort consisted of children treated for complicated SM in 2006–2007 at Queen Elizabeth Central Hospital in Malawi and were followed 7 years later.

### 2.2. Participants

The original cohort recruited 352 SM survivors, 217 sibling controls, and 184 age and sex-matched community controls. In this study, we included children with measured metabolome and hormone profiles from our previous study [18].

### 2.3. Variables

The primary outcome was lean mass index (LMI) 7 years after treatment from severe malnutrition. LMI is the lean mass component of body mass index and is expressed as lean mass by height squared (kg/m^2^). LMI is the reciprocal of impedance data obtained from bioelectrical impedance analysis (BIA) (1/Z) [19]. Independent variables included the metabolome profile and hormone profiles. Age, sex, human immunodeficiency virus (HIV) status, and former nutritional status (case or control from the original trial) were also accounted as confounding factors. The group of control participants for this study included both sibling and community controls.

### 2.4. Data Sources and Measurement

LMI was measured using BIA. As shown previously, the reciprocal of impedance (1/Z) provides a robust marker of LMI without needing population-specific calibration [19]. Targeted metabolomics and hormone profiling were performed using the AbsoluteIDQ^®^ p180 kit (Biocrates, Innsbruck, Austria) and multiplex immunoassay (Milliplex^®^ Mag kits, Millipore, Darmstadt, Germany) as previously described [18].

### 2.5. Study Size

All children with metabolomics, hormone profile, and LMI measurements from our previous study [18] were included in this current secondary analysis. Therefore, sample size was constrained by the availability of prior samples, and no new sample size calculations were done.

### 2.6. Data Analysis

#### 2.6.1. Individual Metabolites and Hormones

First, individual metabolites and hormones were associated with LMI adjusting for age, sex, HIV status, and early childhood nutritional status (case vs. control) using linear models. For metabolites showing significant associations (*p* < 0.05 after false discovery rate (FDR) adjustment), a second model was generated with interaction between early childhood nutritional status and metabolite or hormone. These models were compared using analysis of variance, and the interaction term was retained based on significance of the likelihood ratio (*p* < 0.05). For variables with significant interaction, a stratified analysis was performed to determine individual associations within each group.

#### 2.6.2. Path Analysis

Path analysis is a sub-type of structural equation modeling (SEM) that aims to investigate structural relationships between variables but without the use of latent constructs. We initially generated a hypothetical structural relationship among early-life SM, LMI, and the metabolites or hormones whose associations with LMI were different based on nutritional status (significant interaction between nutrition status and lean mass). Several modifications to this structural relationship were assessed until a valid and acceptable model was achieved. The SEM model was accepted when the following goodness-of-fit criteria were met: root mean squared error of approximation (RMSEA) <0.09, comparative fit index (CFI) >0.9, and Tucker–Lewis index (TLI) >0.9 [20]. SEM was performed using the lavaan package in R.

#### 2.6.3. Predictive Model for Lean Mass Index Using Metabolome and Hormonal Profile

Multivariable predictive models for LMI were generated separately within cases and controls using elastic net regularized regression analysis (EN) [21]. The shrinkage of coefficients of non-contributing variables to zero was controlled by the penalization parameter lambda, which was based on the minimum cross-validated error following ten-fold cross-validation. Several alpha parameter values were assessed, and a final value of 0.5 was taken to achieve a compromise between predictive ability and extracting a fewer number of features. Age, sex, and HIV status were included as covariates. Each EN model was generated using a training set and then validated on held out data using an 80% to 20% split. Models were assessed based on the strength and statistical significance of their correlation (R^2^) and percentage of root mean squared error of prediction (RMSEP). Bootstrapping was performed for 1000 iterations to obtain an optimism-adjusted R^2^ and RMSEP. Analyses were performed in R Version 3.6.

## 3. Results

### 3.1. Study Participants

Metabolomics and hormone profiles from 146 children were used for this analysis after four children (three siblings and one community control) were excluded due to missing LMI measurements from the original case-control study [18] (Figure 1). Participant characteristics are shown in Table 1.

This sub-population reflects the demographics of the entire ChroSAM cohort [4], where SM survivors were observed to have persistent growth deficits compared to controls, specifically lower mid-upper arm circumference, smaller height, and lower body mass and LMI indices. HIV was also more prevalent among SM survivors in our sample population.

### 3.2. LMI Association with Metabolome and Hormone Profile

Adjusting for age, sex, HIV status, and earlier nutritional status (case or control) as potential confounders, LMI was found to be associated with a total of 13 metabolites (including seven phosphatidylcholine (PC) species, two acylcarnitine species (C18 and C18:1), citrulline, tyrosine, tryptophan and creatinine) and two hormones (IGF1 and FGF21) (FDR-adjusted [22] *p* < 0.05) (Table 2; Appendix A). The R^2^ values of all individual models exceed 55%, indicating that individual models comprising of each metabolite or hormone with age, sex, HIV status, and prior nutritional status as covariates explain the majority of the variability in LMI. IGF1 was found to have the highest standardized partial coefficient and partial r^2^, indicating that it has the strongest association with LMI compared to the other metabolites and hormones assessed accounting for age, sex, and HIV status. On the other hand, phosphatidylcholines were the most dominant group among those that showed significant associations with LMI, all of which were significantly negatively associated with lean mass.

Among those with significant association with LMI, we found that the LMI associations with plasma FGF21 and tryptophan had a significant interaction with nutritional status, where significant associations existed in cases but not in controls (Table 2, Figure 2). Cases had significantly higher levels of tryptophan (log mean difference = 0.15, *p* = 0.01) but significantly lower levels of FGF21 (log mean difference = −0.51, *p* = 0.02) than controls. Circulating FGF21 and tryptophan concentrations were significantly negatively correlated (β = −1.21, R^2^ = −0.17, *p* < 0.0001). Due to the significant association between LMI and tryptophan, we further determined the association between LMI and kynurenine/tryptophan ratio. The kynurenine/tryptophan ratio was overall negatively associated with LMI.

### 3.3. Path Analysis

Given the associations between early-life SM, LMI, tryptophan, and FGF21, we explored the possibility that tryptophan and FGF21 mediate the association between early-life SM and lean mass using an SEM approach. Several hypothesis-driven model structures were developed until acceptable goodness-of-fit measures were achieved. Our final model is shown in Figure 3, which converged after 50 iterations. Fit indices are: RMSEA = 0.06, CFI = 0.99, TLI = 0.95, which indicates that the model achieved a good fit [20]. The variance explained by the model for LMI is 61.7%, tryptophan 6.6%, and FGF21 18.8%.

The model suggests a direct negative association between early-life SM and LMI, with this association partially mediated by plasma tryptophan levels. Early-life SM has a negative impact on plasma tryptophan levels, whereas tryptophan has a significant positive association with lean mass. Furthermore, the association between early-life SM and FGF21 is fully mediated by tryptophan. Moreover, the SEM suggests that a direct association between FGF21 and lean mass is not statistically significant.

### 3.4. Prediction of Lean Mass Index Using Metabolome and Hormonal Profiles

To reveal potential differences in LMI associations with metabolome and hormones between cases and controls, we generated separate multivariable models for each group using EN. The EN models predicted LMI of controls with high accuracy and robustness with R^2^ > 0.7 and percentage of root mean squared error of prediction (RMSEP) ≤15% of mean LMI (Figure 4). However, the model built for cases was not as strongly predictive with much lower R^2^ and higher RMSEP values. The EN models retained nine variables for the control group and 14 for the case group and only age, IGF1, and FGF23 were selected as predictors in both groups. Predictors for LMI among controls were dominated by lipid species, which showed an overall negative relationship with LMI, whereas the main features for cases were FGF21, amino acids, including tryptophan, and their metabolites.

## 4. Discussion

In this secondary analysis, we demonstrated that circulating hormones and metabolites were associated overall with LMI among children in Malawi. However, the associations of tryptophan and FGF21 toward LMI were only statistically significant among children with a history of SM, whereas these associations were not observed among controls. This indicates a potential influence of early-life malnutrition on growth signaling, especially lean mass accretion as influenced by protein or amino acid availability. This study builds on previous ChroSAM study findings reporting that children who experienced SM in early childhood were still smaller and had less lean mass, even 7 years post treatment [4].

Results from SEM indicate that the association between early-life SM and LMI was mostly mediated by plasma tryptophan levels. This observation could be explained by SM in early life impacting the metabolome, especially tryptophan metabolism, which then has a causal effect on lean mass. A previous study showed that a tryptophan-deficient diet reduced the skeletal mass in mice [23]. In addition, an earlier study in pigs revealed that tryptophan deficiency led to a reduced protein synthesis rate in muscles, which recovered upon higher tryptophan intake [24]. This observation concurred with an even earlier study indicating that dietary tryptophan improved the incorporation of ^13^C phenylalanine into muscle protein in pigs, indicating that tryptophan improved protein synthesis beyond its incorporation to protein [25]. These studies demonstrate a potential direct causal relationship between tryptophan and lean mass accretion. However, these studies focused on muscle mass, which is only one component of lean mass. Other contributors to lean mass are bones, organs, and other fat-free components.

In observational clinical studies, serum tryptophan concentration was found to be positively associated with lean mass in cancer patients with skeletal muscle atrophy [23]. Furthermore, tryptophan and lysine were positively associated with appendicular lean mass but not with other adiposity-related measures in older black men [26].

The role of early-life SM in regulating tryptophan metabolism remains to be fully elucidated by mechanistic studies. For instance, it would be interesting to investigate whether lower tryptophan levels are due to a persistent downregulation of amino acid transporters in the intestines (i.e., sodium-dependent neutral amino acid transporter B(0)AT1) or whether an increase in tryptophan metabolism (enhanced activity of intestinal indoleamine 2,3-dioxygenase or hepatic tryptophan 2,3-dioxygenase) in response to low-grade inflammation persists in children with SM later in life.

FGF21, on the other hand, is increased during dietary restriction [27] or protein malabsorption or deficiency [28,29]. A study in mice previously showed that tryptophan restriction alone partially recapitulates the metabolic effect of total amino acid restriction, including FGF21 signaling [30]. Our data concurs with these previous observations on the negative relationship between FGF21 and tryptophan, which was independent of malnutrition in early life. FGF21 is an endocrine signal of protein restriction [31]. Taking these markers together, protein or amino acid availability seems to be an underlying determinant of reduced LMI among SM survivors. However, we did not find evidence that the association between early-life SM and lean mass is mediated by FGF21, nor that the association between FGF21 and LMI is independent of tryptophan levels. Hence, it is likely that the link we observed between FGF21 and LMI was due largely to the significant correlation between tryptophan and FGF21.

Our multivariable prediction models showed that model fit was better for controls (validation R^2^ = 0.75) than cases (validation R^2^ = 0.52). This indicates that more unmeasured factors could explain the reduced lean mass accretion among SM survivors compared to children with no previous exposure to SM. In future studies, it will be useful to investigate these underlying factors to fully understand the long-term consequences of poor early-life nutrition.

Amino acids and amino acid metabolites dominated the top predictors for LMI among cases. Increased circulating levels of essential amino acids (EAA) such as tryptophan, threonine, and methionine, and reduced levels of spermine and methionine sulfoxide were shown to be associated with increased LMI. Indeed, a recent study showed that a deprivation of threonine and tryptophan drives the systemic metabolic response of total amino acid deprivation and dietary protein depletion [32]. Therefore, these EAA are limiting amino acids, and their availability from the diet or the body’s absorptive capacity of these EAA (down-regulation of amino acid transport mechanisms) could strongly influence LMI.

Taken together, our findings indicate protein or amino acid availability as a potentially modifiable pathway to improve the health outcomes of SM survivors. Therefore, studies to understand the role of early-life SM on amino acid metabolism, especially tryptophan, are necessary.

In this study, a limitation is that lean mass was estimated based on BIA rather than measured with more direct quantification methods such as MRI, the isotope method, or dual-energy X-ray absorptiometry. However, the form of data expression we used here is independent of population differences in physique, and it has been validated against criterion methods, demonstrating that it is robust for ranking individuals within any population [19,33]. Moreover, this is a secondary analysis of data obtained from a case-control study investigating metabolomics and hormonal differences between SM survivors and controls [18], and therefore, it was not specifically designed with LMI as an outcome of interest. We also recognize the high prevalence of HIV, a disease known to affect lean mass, within this study population (15% HIV positive, 22% unknown HIV status), especially among SM survivors (27% HIV positive compared to 5% HIV positive among controls). However, we did include HIV as a confounding factor in both univariable and multivariable analyses to reduce this potential bias. This follow-up cohort was also recruited from a randomized control trial [34], where more than half of the children had oedematous malnutrition. Although we were not able to find evidence of any significant association between prior SM phenotype (i.e., oedematous SAM or severe wasting) on the metabolome and hormone profile [18], its potential long-term effect could not be discounted. Lastly, as this is a follow-up cohort after 7 years, survivor bias could influence the associations we observed in this study. Therefore, validation of these results in an independent cohort is warranted. Prospective population-based cohorts would be especially helpful in understanding mechanisms since our data focus on those with a specific type of SM who were sick enough to be admitted to an inpatient feeding center. In contrast, today’s programs aim to be more proactive and identify children in the community when less severely malnourished: it would be important to know if such approaches also result in less long-term metabolic impairment. This would add weight to calls for early treatment and prevention.

Nevertheless, we present some of the first physiological data on the long-term outcomes of early-life malnutrition on lean mass. Another strength of this work is that the cohort in this study was recruited from a clinical trial. This means that data collection was standardized and the clinical histories of the children are available, which ensures that cases are well defined contrary to studies relying on memory recall or retrospective data.

## 5. Conclusions

Early-life SM is associated with persistent effects on metabolic processes later in life, which may influence lean mass in survivors of childhood SAM. The association between early-life SM and lean mass could be mediated by plasma tryptophan concentrations, implicating the role of tryptophan metabolism in lean mass accretion among survivors of SM. These findings provide more potential mechanistic understanding for the increased NCD risk observed in SM survivors later in life.

## Figures and Tables

**Figure 1 nutrients-12-03593-f001:**
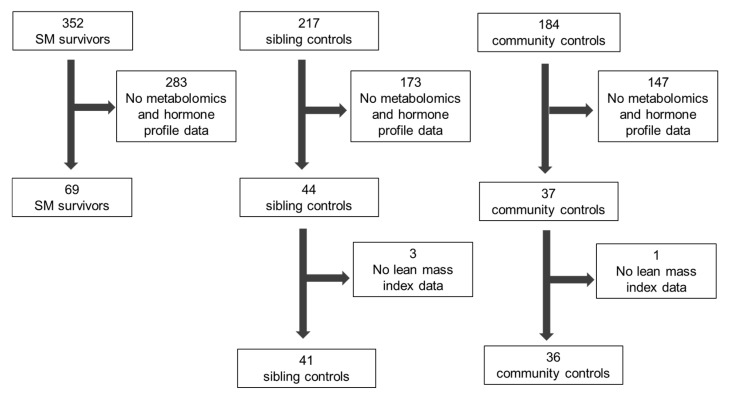
Recruitment flow diagram.

**Figure 2 nutrients-12-03593-f002:**
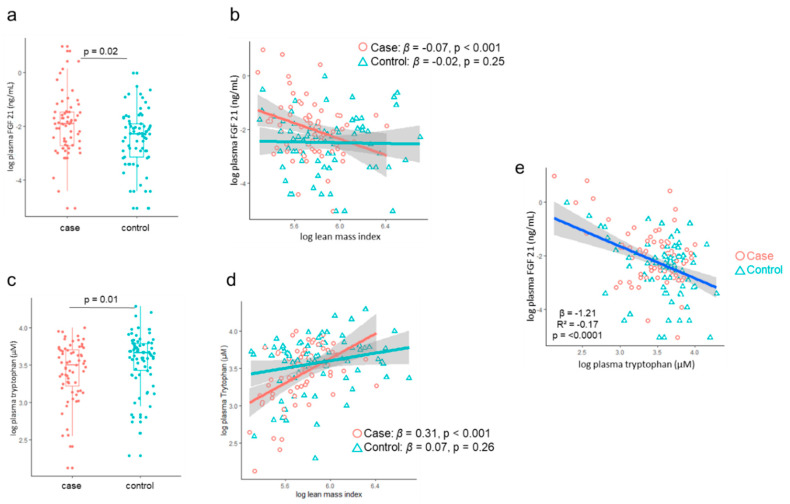
FGF21 and tryptophan levels and their association with lean mass. Concentration differences in plasma FGF21 (**a**) and tryptophan (**c**) in survivors of childhood malnutrition (case) and sibling and community controls in the ChroSAM cohort. Association between lean mass index and FGF21 (**b**) and tryptophan (**d**) stratified by early childhood malnutrition status adjusted for age, sex, and HIV status. The associations between lean and tryptophan (*p* interaction = 0.003) and FGF21 (*p* interaction = 0.007) were significantly modified by early-life malnutrition. (**e**) Correlation between plasma FGF21 and tryptophan. No evidence of interaction between early malnutrition and the association between FGF21 and tryptophan was found (*p* = 0.63).

**Figure 3 nutrients-12-03593-f003:**
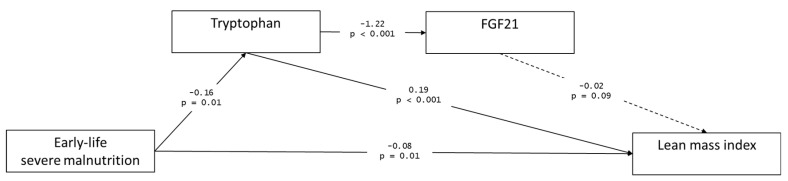
Path analysis of the associations among early-life severe malnutrition (SM), lean mass index, plasma tryptophan and plasma FGF21. Goodness-of-fit indices: root mean squared error of approximation (RMSEA) = 0.06, comparative fit index (CFI) = 0.99, Tucker–Lewis index (TLI) = 0.95. Full arrows are statistically significant (*p* < 0.05), broken arrow is not statistically significant. Values of arrow represent the association estimate β above and the *p*-value below. All paths are controlled for age, sex, and HIV status. Full statistical estimates can be found in Appendix A.

**Figure 4 nutrients-12-03593-f004:**
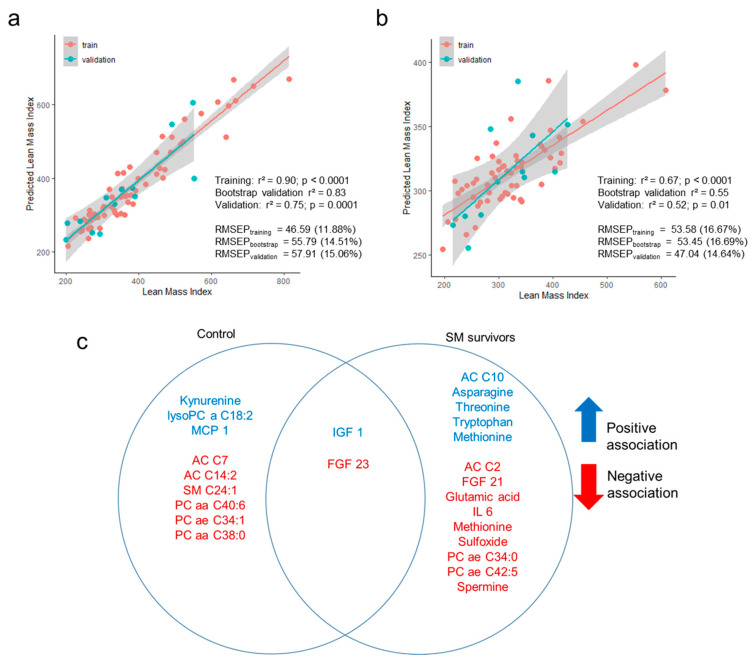
Elastic net regularized regression model prediction of lean mass index. Independent variables included age, sex, HIV status, 155 plasma metabolite concentrations, and 22 hormones. Correlation between observed lean mass index and predicted lean mass index among control (**a**) and SM survivors (**b**). Training set included 80% of the observations and the remaining 20% was used for external validation. RMSEP = root mean squared error of prediction, the numbers in parentheses indicate the percentage of mean lean mass index. Bootstrap validation of the training set was performed using 1000 iterations. (**c**) Metabolite and hormone features selected by the elastic net model. Features in blue are positively associated with lean mass while features in red are negatively associated.

**Table 1 nutrients-12-03593-t001:** Characteristics of study participants in this secondary metabolomics analysis.

	SM Survivors	Controls	Mean Difference °	*p*
*n*	69	77		
Mean age in years (SD) *	9.63 (1.63)	9.95 (2.31)	−0.32	0.33
*n* girls (%)	30 (43%)	34 (44%)		1
*n* HIV status (%)				<0.001
Negative	44 (64%)	46 (60%)		
Positive	20 (29%)	4 (5%)		
Unknown	5 (7%)	27 (35%)		
Mean mid-upper arm circumference in cm (SD) *	17.12 (1.64)	18.15(2.30)	−1.03	0.002
Mean height in cm (SD) *	125.51 (8.90)	129.57 (13.73)	−4.06	0.038
Mean length-for-height z-score (SD) *	−1.68 (1.21)	−1.37 (1.00)	−0.31	0.090
Mean weight-for-age z-score (SD) *	−1.44 (0.93)	−1.2 (0.88)	−0.24	0.210
Mean body-mass index in kg/m^2^ (SD) *	15.25 (1.33)	15.77 (1.60)	−0.52	0.037
Mean lean mass index in (1/Z) × 1000 (SD) *	319.97 (75.36)	378.87 (132.61)	−58.9	0.001

° SM survivors—controls, * calculated from the sample population of study.

**Table 2 nutrients-12-03593-t002:** Metabolites and hormones significantly associated with lean mass index.

Feature	Partial Coefficient *	*p* °	Partial r^2^ ^□^	R^2^ ▪	*p °* Interaction ^§^
*Hormones*					
IGF1	0.125	<0.001	0.271	0.667	-
FGF21	−0.054	0.027	0.069	0.574	0.007
*Lipids*					
AC C18	−0.054	0.021	0.073	0.576	-
AC C18:1	−0.050	0.046	0.058	0.569	-
PC ae C34:1	−0.065	0.004	0.107	0.592	-
PC aa C32:0	−0.060	0.005	0.095	0.586	-
PC aa C34:1	−0.062	0.006	0.096	0.587	-
PC aa C36:1	−0.064	0.006	0.095	0.586	-
PC aa C32:1	−0.058	0.012	0.082	0.580	-
PC ae C34:0	−0.051	0.027	0.068	0.574	-
PC ae C38:5	−0.052	0.027	0.067	0.573	-
*Amino acids*					
Tryptophan	0.079	<0.001	0.150	0.611	0.003
Tyrosine	0.057	0.012	0.082	0.580	-
Citrulline	0.052	0.040	0.061	0.571	-
*Biogenic amine*					
Creatinine	0.064	0.004	0.100	0.589	-
*Ratio and sums*					
Kynurenine:Tryptophan	−0.07	<0.001	0.003	0.590	-
Sum of aromatic amino acids	0.048	0.040	0.059	0.569	-
Sum of lysine, threonine and tryptophan	0.073	0.001	0.122	0.598	0.03

* Standardized regression coefficient adjusted for age, sex, and HIV status; positive values indicate a positive association while negative values indicate a negative association, ^□^ Coefficient of partial determination, indicating the variation in lean mass index (LMI) that each metabolite or hormone explains independent of the other covariates (age, sex, HIV status, and early-life severe malnutrition), ▪ Coefficient of multiple determination (R^2^) of the full multivariate model including age, sex, HIV status, and early-life severe malnutrition as covariates, ° adjusted for false discovery rate (FDR), ^§^ interaction between feature and early childhood malnutrition status were tested and only further analyzed if the likelihood ratio test suggested an improved model fit (*p* < 0.05).

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
