# Peer review of "Childhood Malnutrition and Association of Lean Mass with Metabolome and Hormone Profile in Later Life"

_nutrients, 2020, doi:10.3390/nu12113593_

Round 1

Reviewer 1 Report

I have carefully reviewed the manuscript nutrients-1004398 with Dr. Gerard Bryan Gonzales as the first author. The title is " Childhood malnutrition and association of lean mass with metabolome and hormone profile in later life”. The purpose of this study is to explore factors associated with lean mass deficit in early-life severe malnutrition survivors by assessing its association with the metabolome and hormonal profile among Malawian children.

What does seem sure is tackling severe malnutrition is a major global health priority. This research area is very important for child health. The clinical study [Ref.4] seems to have been adequately conducted according to the protocol and the results are clearly presented. This study was conducted as the secondary analysis. It contains enough interest and originality to merit publication. Prior to publication, authors should add unit of each parameter in Table 1.

Author Response

We thank the reviewer for his/her positive evaluation of our manuscript. We now have included units on all parameters in Table 1. See line 134-135

Reviewer 2 Report

This is a good study. I have a few comments and suggestions.

  1. It was not clear to me from the manuscript when the blood samples were taken. Were they taken during the period of malnutrition? If not, that changes the interpretation of the data substantially. Rather than citing the former studies, perhaps give a bit more detail of the experiment to the reader.
  2. Several amino acids (i.e. Met, Val etc..) were not reported. Why was this? Were they not measured or not measured properly? In any case, it would be good to mention this as a limitation as you are not covering all of the amino acids which could contribute to this response. This is particularly important, as studies in rodents have shown that Met or Val restriction can induce changes in body composition in mice (PMID: 29266268, PMID: 32012481).
  3. A recent study (https://www.nature.com/articles/s41467-020-16568-z ) has shown that dietary essential amino acids mediate the majority of the systemic metabolic remodelling with dietary protein restriction. This paper, along with others (PMID: 26330054, PMID: 22374976, PMID: 27548521, PMID: 28752051) on this topic should be cited as they examine relationships between protein and amino acid nutrition, FGF21 and body composition as done here. Regarding the aforementioned study, the authors suggested that the dietary restriction of a certain class of essential amino acid, the metabolically essential amino acids LYS, THR, and TRP, may be more important for effects of total protein restriction than any other AA. Given that not every child will have the same foodstuffs, and thus may have access to different limiting AA, then perhaps some relationships between LMI, FGF21, and different ratios or summed AAs is warranted. For example, one could sum KTW and the other EAA (FHIL) and see which of these has a stronger relationship to LMI and FGF21.
  4. On FGF21, could the LMI be a result of FGF21 based upon the papers cited in the manuscript as well as those mentioned above which have examined the role of FGF21?
  5. It is noted that data from both boys and girls were included. Since the effects of diet are known to be different between the sexes in rodents perhaps this could be examined as well?
  6. Is there any capacity to examine other hormones which might contribute here such as GDF15 and follistatin?

Author Response

1. The blood samples were taken 7 years after the children have been treated for malnutrition. We believe that we have interpreted our results appropriately. In this manuscript, we showed associations between lean mass index with their metabolome and hormone profile 7 years after treatment from malnutrition, and differential associations between those who were previously malnourished and controls. We have now added more details on the analysis in lines 67-71

2. We thank the reviewer for this very important observation. We used the Biocrates p180 kit to measure amino acids, biogenic amines and lipids. During data cleaning, while removing variables that are not necessary for the analysis (demographic data, etc), we accidentally removed 4 extra columns, which were methionine, arginine, valine and creatinine. Our coverage of amino acids includes: Alanine, Arginine, Asparagine, Aspartate, Citrulline, Glutamate, Glutamine, Glycine, Histidine, Isoleucine, Leucine, Lysine, Methionine, Ornithine, Phenylalanine, Proline, Serine, Threonine, Tryptophan, Tyrosine, and Valine, which we have now updated in the supplementary table 1. We apologize for this oversight. This however did not change the overall conclusion of the study.

Upon revision of the statistics, we found that neither valine nor methionine were significantly associated with lean mass index in our population. However, the updated multivariable model for LMI now extracted methionine as one of the positive predictors (Figure 4 now updated). The main model characteristics did not change, except for an increase in validation R² to 0.52 from 0.47 with the inclusion of methionine.

The discrepancy in results with mice studies could be explained by the fact that these animal models studied the effect of amino acids in the context of obesity. None of the children followed in this cohort were obese. Also, as we explained in lines 286-298, our cohort included children with history of oedematous malnutrition and HIV, which may have influenced these associations.

3. We thank the reviewer for recommending this reference, which indeed is very important to this manuscript. We have now cited it and another of the recommendations. See lines 267-277.

We have now included some summary results, including total EAA, total aromatic AA, KTW and FHIL. Except for KTW, none of the sums were associated with LMI. KTW was found to be significantly associated with LMI, and its association had a significant interaction with early life malnutrition. However, as we now show in Table 2, this association is not superior to TRP alone, indicating that TRP could be the main driver of this association. We also performed a path analysis using KTW instead of TRP, and we observed that the model statistics were unstable and did not pass validation criteria: RMSEA = 0.112 (cut-off < 0.08), TLI = 0.80 (cut-off > 0.9), CFI = 0.98 (cut-off > 0.9). This indicates that TRP alone is better at explaining the mediation of the association between early-life malnutrition and LMI than KTW.

4. Our results indeed suggest that FGF21 is negatively association with LMI, as we have discussed in the paper. However, we found that this association is fully mediated by plasma tryptophan concentrations. We therefore believe that in this particular population, the negative association between FGF21 and LMI could be due to the strong negative association between tryptophan and FGF21. Tryptophan appears to have a stronger association with LMI than FGF21.

Zapata et al (2019) showed that tryptophan restriction was associated with metabolic changes in mice, including influencing FGF21 signalling. This observation shows that if tryptophan influences FGF21 and lean mass, that the associations between FGF21 and LMI could be via tryptophan.

5. Our analyses were adjusted for sex. However, the main focus of our paper is to show the association between early-life malnutrition and future LMI. Stratifying our population to determine sex effects would reduce the power of our analysis as our study design did not take this stratification into account. We will keep this in mind for our future studies.

6. We thank the reviewer for his/her suggestion. We are in the process of securing funding to allow us to continue this study, including follow-up of the participants and comparing these results to malnutrition survivors in other countries. We will definitely keep it in mind to assess GDF15 and follistatin.

Reviewer 3 Report

The manuscript entitled “Childhood malnutrition and association of lean mass with metabolism and hormone profile in later life” (nutrients-1004398), presented by Gonzales et al. is well designed, the methodological approaches used are appropriate and also is well written.

Author Response

We would like to thank the reviewer for his/her positive feedback on the manuscript. No changes were made based on the reviewer’s comment.

Reviewer 4 Report

Dear Authors,

This is a very interesting research topic. Introduction lays out a clear picture on importance of the study. Methods, including the statistical analysis, results and discussion sections are also well written. The findings are easy to follow as they are aligned step by step with the analysis plan. I have a few very minor comments.

Results:

  1. Line 153 and 154: probably a typo. It should be “cases” at the beginning of the sentence. "Controls had significantly higher levels of tryptophan (log mean difference = 0.15, p = 0.01) but significantly lower levels of FGF21 (log 153 mean difference = -0.51, p = 0.02) than controls."

This typo affects readers’ understanding of the findings later on the manuscript.

Discussion:

Some minor suggestions:

  1. What about untargeted metabolomics approach? As this is a previously collected data/samples, authors are limited in their investigation and data analysis. However, knowing what the authors think about using an untargeted Metabolomic approach in future toward the understanding of metabolomic pathways in severe malnutrition cases versus control would be helpful? How would this change the knowledge base related to the topic?
  2. It would be interesting to hear authors opinions on how the findings of this study influence the existing prevention efforts focused toward improving lean body index in severe malnutrition cases from the clinical perspective.

Author Response

1. We have now corrected this typo by changing “Controls” to “Cases” in line 155

2. We thank the reviewer for his/her suggestion. We are in the process of securing funding to allow us to continue this study, including follow-up of the participants and comparing these results to malnutrition survivors in other countries. In our follow-up study, we will perform untargeted metabolomics and lipidomics to get a broader coverage of metabolites and pathways to be investigated.

3. Our current study, along with our previous report on the ChroSAM cohort, highlights the long-term effects of severe malnutrition. Hence, efforts to prevent severe malnutrition in children is the best way to ensure reduce this risk. This is addressed in lines 301-304.